# Longitudinal assessment of tumor development using cancer avatars derived from genetically engineered pluripotent stem cells

Tomoyuki Koga [1,2,9], Isaac A. Chaim [3,5,9], Jorge A. Benitez [1], Sebastian Markmiller[3], Alison D. Parisian[1,4], Robert F. Hevner[4], Kristen M. Turner[1], Florian M. Hessenauer[1], Matteo D'Antonio [5], Nam-phuong D. Nguyen[6], Shahram Saberi[7], Jianhui Ma[1], Shunichiro Miki[1], Antonia D. Boyer[1], John Ravits[7], Kelly A. Frazer [5,8], Vineet Bafna[6], Clark C. Chen[2], Paul S. Mischel [1,4], Gene W. Yeo[3,5*] & Frank B. Furnari [1,4*]

Many cellular models aimed at elucidating cancer biology do not recapitulate pathobiology including tumor heterogeneity, an inherent feature of cancer that underlies treatment resistance. Here we introduce a cancer modeling paradigm using genetically engineered human pluripotent stem cells (hiPSCs) that captures authentic cancer pathobiology. Orthotopic engraftment of the neural progenitor cells derived from hiPSCs that have been genome-edited to contain tumor-associated genetic driver mutations revealed by The Cancer Genome Atlas project for glioblastoma (GBM) results in formation of high-grade gliomas. Similar to patient-derived GBM, these models harbor inter-tumor heterogeneity resembling different GBM molecular subtypes, intra-tumor heterogeneity, and extrachromosomal DNA amplification. Re-engraftment of these primary tumor neurospheres generates secondary tumors with features characteristic of patient samples and present mutation-dependent patterns of tumor evolution. These cancer avatar models provide a platform for comprehensive longitudinal assessment of human tumor development as governed by molecular subtype mutations and lineage-restricted differentiation.

[1] Ludwig Cancer Research San Diego Branch, 9500 Gilman Dr., CMM-East Room 3055, La Jolla, CA 92093, USA. [2] Department of Neurosurgery, University of Minnesota, 420 Delaware St SE, Minneapolis, MN 55455, USA. [3] Department of Cellular and Molecular Medicine, University of California San Diego, 2880 Torrey Pines Scenic Drive, La Jolla, CA 92093, USA. [4] Department of Pathology, University of California San Diego, 9500 Gilman Dr., La Jolla, CA 92093, USA. [5] Institute for Genomic Medicine, University of California San Diego, 9500 Gilman Dr. Mail Code 0761, La Jolla, CA 92093, USA. [6] Department of Computer Science and Engineering, University of California San Diego, 9500 Gilman Dr., Mail Code 0404, La Jolla, CA 92093, USA. [7] Department of Neuroscience, University of California San Diego, 9500 Gilman Dr., Mail Code 0662, La Jolla, CA 92093, USA. [8] Department of Pediatrics and Rady Children's Hospital, University of California San Diego, 9500 Gilman Dr., Mail Code 0831, La Jolla, CA 92093, USA. [9] These authors contributed equally: Tomoyuki Koga, Isaac A. Chaim. *email: geneyeo@ucsd.edu; ffurnari@ucsd.edu

Effective modeling of cancer has been a conceptual cornerstone in the field of oncology for studying pathobiology and identifying therapeutic targets. In the case of glioblastoma (GBM), the most common primary malignant tumor of the central nervous system[1], mouse models of GBM-like tumors generated through the genetic disruption of different combinations of core tumor suppressors and/or by introduction of oncogenes such as *Src*, *K-ras*, *H-ras*, *PDGFB*, and *EGFRvIII*[2] are available to investigate the biology of these aggressive tumors or to test possible treatments in preclinical settings[3]. These mouse models are suitable to investigate pathology of genetically defined gliomas and useful for drug testing, but typically lack the intratumor heterogeneity that is observed in human gliomas[4]. In addition, while human astrocytes engineered with combinations of human *TERT* and *H-Ras* expression and inhibition of the TP53 pathway either by SV40 T/t-Ag or by HPV E6 and E7 generate gliomas with high-grade histology[5,6], how well these models recapitulate the full spectrum of glioma pathobiology, especially in terms of GBM heterogeneity, has not been well defined. In contrast, patient-derived xenografts (PDX) have been useful to study inter- and intra-tumoral heterogeneity[7,8] and sensitivity to pathway-specific therapies[9], however, they do not allow for experimental standardization or afford analysis of the effects of molecular subtype mutations on tumor evolution.

The progress in human stem-cell technologies and genome editing using site-specific nucleases such as ZFN, TALEN, and CRISPR/Cas9 has broadened the field of human disease modeling[10]. Such engineering has also been efficiently applied to neural stem cells providing opportunities for functional genetic analysis[11]. This combination of human stem cell and genome editing promises great potential when applied to cancer models. The first such model generated utilized colon organoids derived from human intestinal crypt stem cells engineered with four or five mutations common in colorectal cancers[12,13]. These organoid models accurately predict drug responses and their utility is anticipated for application of personalized therapies[14]. Later, a brain tumor model deleted for *PTEN* by TALEN-mediated homologous recombination led to the reprograming of human neural stem cells toward a cancer stem cell-like phenotype[15]. However, it remains unknown if these cancer models generated through genome editing harbor authentic pathological features of cancers, including tumor heterogeneity and clonal evolution.

Here, we establish a robust platform in an isogenic background, which uses CRISPR/Cas9 genome editing technology and serial in vivo engraftments enabling longitudinal assessment of human high-grade glioma (HGG) models containing combinations of genetic alterations observed in proneural and mesenchymal GBM molecular subtypes. We further present how closely these models recapitulate pathobiology of the disease and discuss their utility as an avatar platform for future studies on tumor biology and evolution.

## Results

**Neural progenitors with GBM mutations form HGG-like tumors.** We first introduced two different combinations of driver mutations into human induced pluripotent stem cells (iPSCs) by CRISPR/Cas9 genome editing[16,17] (Fig. 1a, b). One combination of deletions targeted tumor suppressor genes *PTEN* and *NF1*, which are commonly altered together in the mesenchymal subtype of GBM[18,19]. A second combination of deletions targeted *TP53* and exons 8 and 9 of *PDGFRA* (*PDGFRA*$^{\Delta 8-9}$). This creates a constitutively active truncating PDGFRA mutation observed in 40% of *PDGFRA* amplified GBM[20], resulting in a genotype commonly found in the proneural subtype of isocitrate dehydrogenase-wildtype GBM[18,19]. The genetic modifications in

single clones were confirmed by genotyping PCR (Fig. 1c) and RT-qPCR (Fig. 1d). Edited iPSC clones with desired mutations were differentiated into neural progenitor cells (NPCs), using a small molecule protocol[21] and differentiation status was confirmed by downregulation of pluripotency markers, Nanog and Oct4, and corresponding upregulation of NPC markers, Pax6, Nestin, and Sox1 (Fig. 1e). These edited NPCs were expanded on matrigel-coated plates in NPC maintenance media[21] and were utilized in further experiments.

We next evaluated if these genetically modified NPCs were capable of forming orthotopic tumors in immunocompromised mice (Fig. 1a). When edited NPCs were engrafted in the brains of four Nod *scid* mice, *PTEN*$^{-/-}$;*NF1*$^{-/-}$ NPCs and *TP53*$^{-/-}$; *PDGFRA*$^{\Delta 8-9}$ NPCs each formed brain tumors with median survival of 141, and 119.5 days, respectively (Fig. 1f). Pathological assessment of *PTEN*$^{-/-}$;*NF1*$^{-/-}$ tumors revealed regions of hypercellularity with occasional mitoses (Fig. 2a), and in one out of four tumors, there were biphasic dense glial and loose mesenchymal/sarcoma morphologies, typical of gliosarcoma (Fig. 2b). In addition, regions of necrosis (Fig. 2c), vascular endothelial proliferation (Fig. 2d), subarachnoid spread (Fig. 2e), perineuronal satellitosis, and subpial accumulation of tumor cells (Fig. 2f), were also apparent. The tumors were consistently positive for GFAP (3+ in 6/6 high power fields) (Fig. 2g, Supplementary Fig. 2a) and Olig2 (3+ in 4/6 high power fields, 2+ in 2/6 high power fields) (Fig. 2h, Supplementary Fig. 2b), and highly proliferative as indicated by Ki-67 staining (35.44 ± 1.435%; mean ± SEM) (Fig. 2i, Supplementary Fig. 2c). *TP53*$^{-/-}$; *PDGFRA*$^{\Delta 8-9}$ tumors presented nodular growth of a primitive neuronal component (dark purple) intermingled with glial components (Fig. 2j), rosettes with neuropil-like texture (Fig. 2k), a serpiginous zone of pseudopalisading necrosis (Fig. 2l), and intraventricular growth (Fig. 2m). These tumors were positive for GFAP (3+ in 6/6) (Fig. 2n, Supplementary Fig. 2a) and Olig2 (3+ in 6/6) (Fig. 2o, Supplementary Fig. 2b), and also highly positive for Ki-67 staining (27.79 ± 5.731%; mean ± SEM) (Fig. 2p, Supplementary Fig. 2c). In terms of WHO grade, three and one out of four tumors were scored as grade 4 and grade 3, respectively for *PTEN*$^{-/-}$;*NF1*$^{-/-}$ tumors, and four out of four tumors were scored as grade 4 for *TP53*$^{-/-}$;*PDGFRA*$^{\Delta 8-9}$ tumors (Supplementary Fig. 3). In contrast, *PTEN*$^{-/-}$ and *TP53*$^{-/-}$ singly edited NPCs did not form tumors in the brain over the same time span (Fig. 1f, Supplementary Fig. 4a, b), while unedited iPSCs formed teratoma-like tumors (Supplementary Fig. 5). Lack of teratomas after NPC injection suggests a high efficiency of differentiation to NPCs. These results illustrate that using this modeling paradigm, small numbers of known driver mutations found in GBM are sufficient for phenotypic recapitulation of human HGG tumors.

**iHGG cells can be cultured and form secondary tumors.** One of the benefits of using PDX models in cancer research is that they can be cultured in vitro and be re-engrafted in animals, thus enabling both in vitro and in vivo analyses[22]. We evaluated if our induced HGG (iHGG) models could be used in a similar manner. Dissociated tumors obtained from the mouse brains were sorted for human cells using a human MHC antibody, followed by propagation of isolated cells in the same neurosphere conditions used for GBM PDX spheres[23], which confirmed iHGG sphere formation capability (Fig. 3a). These iHGG spheres possessed the same genotypes as the corresponding input NPCs (Supplementary Fig. 6). Extreme limiting dilution assays[24] showed that iHGG spheres had greater self-renewal capacity, a feature of cancer stem cells, when compared to pre-engraftment NPCs (Fig. 3b), again

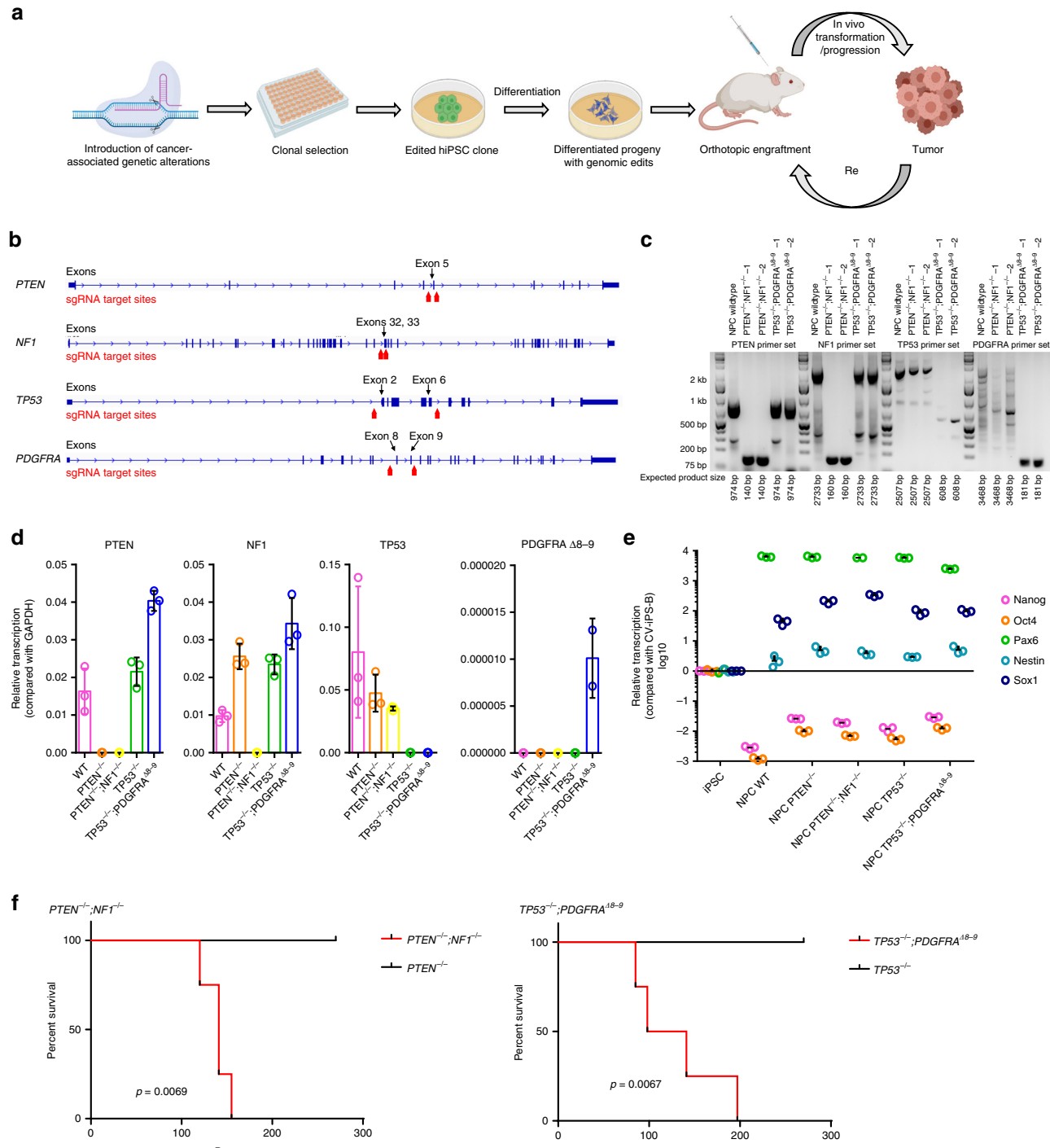

**Fig. 1 Different iHGG models derived from edited human iPSCs. a** Schema of iHGG generation. **b** Designs for gene editing indicating placement of sgRNAs. **c** Genotyping PCR and **d** Semi-quantitative RT-qPCR evaluating designated edits. Data are representative of three replicates, $n = 3$. Data are represented as mean ± SD. **e** RT-qPCR results of markers for iPSCs and NPCs. Data are representative of three replicates, $n = 3$. Data are represented as mean ± SD. **f** Kaplan–Meier curves showing survival of mice engrafted with (left) $PTEN^{-/-}$ NPCs, $PTEN^{-/-};NF1^{-/-}$ NPCs, (right) $TP53^{-/-}$ NPCs, and $TP53^{-/-};PDGFRA^{\Delta 8-9}$ NPCs. Statistical significance was evaluated by the log-rank test. $n = 4$ animals for each arm for each model. Source data are provided as a Source Data file.

highlighting gain of cancerous phenotypes of iHGG cells compared to original input cells.

We then evaluated if these iHGG-derived sphere cells maintained tumorigenic capacity by secondary orthotopic engraftment (Fig. 3c). When injected in the brains of Nod *scid* mice, $PTEN^{-/-};NF1^{-/-}$ and $TP53^{-/-};PDGFRA^{\Delta 8-9}$ iHGG-derived sphere cells formed tumors with a shortened latency period of median 76.5 days and

34.5 days, respectively ($p = 0.0005$, log-rank test) (Supplementary Fig. 7). We also tested if these models can be used for in vivo drug treatment experiments comparable to those applied to PDX lines by treating orthotopically engrafted animals with temozolomide (TMZ), a DNA-alkylating chemotherapeutic agent used for standard care treatment of GBM patients[25]. $TP53^{-/-};PDGFRA^{\Delta 8-9}$ iHGGs proved to be more sensitive to TMZ compared to $PTEN^{-/-};NF1^{-/-}$

PTEN$^{-/-}$;NF1$^{-/-}$

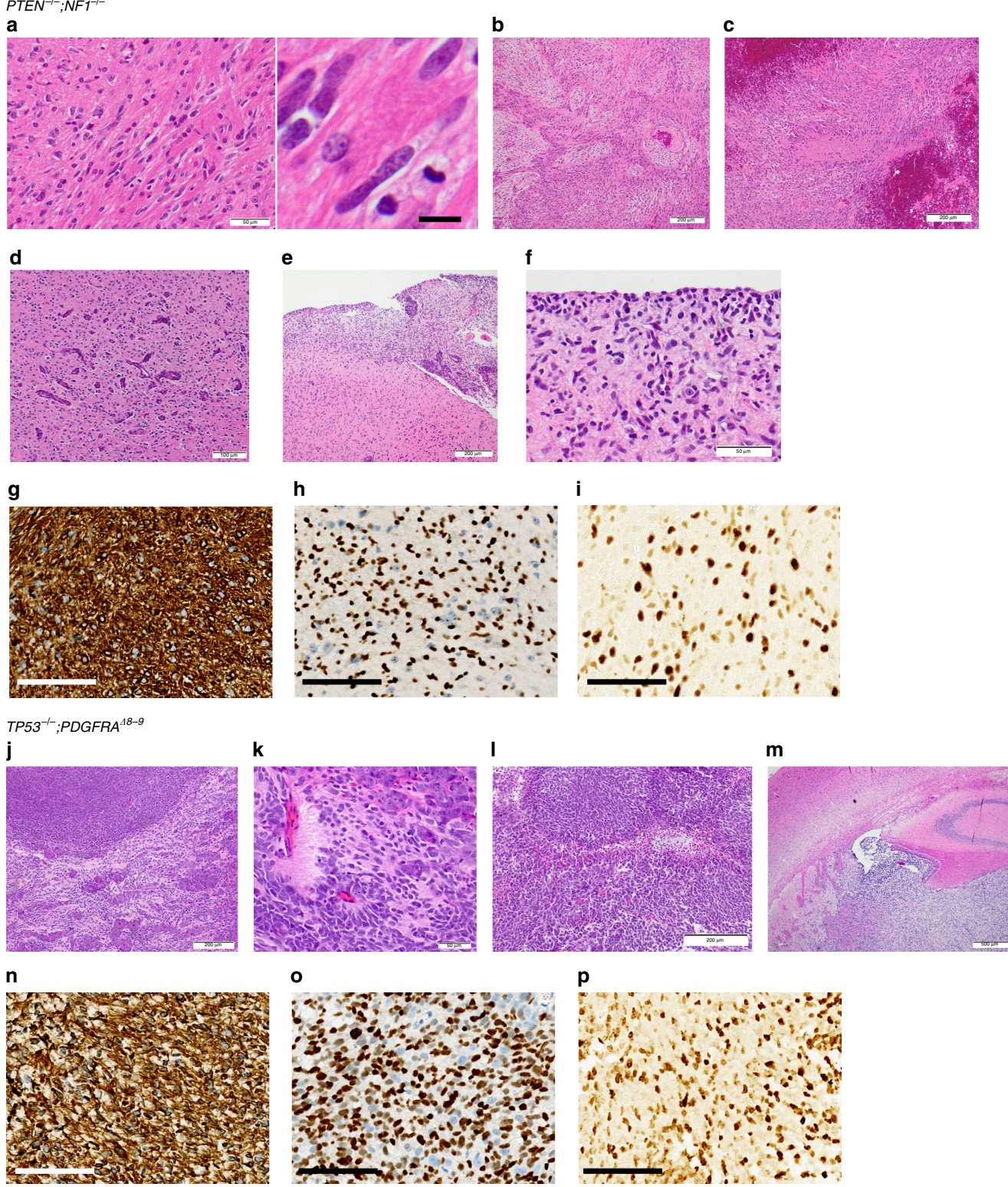

TP53$^{-/-}$;PDGFRA$^{\Delta 8-9}$

iHGGs (Fig. 3d). PTEN$^{-/-}$;NF1$^{-/-}$ iHGGs were found to express higher levels of $O^6$-methylguanine DNA methyl transferase (MGMT) (Fig. 3e), which is associated with resistance to TMZ in GBM patients[26], compared to TP53$^{-/-}$;PDGFRA$^{\Delta 8-9}$ iHGGs. An alternative explanation of this differential sensitivity, through MGMT-independent mechanisms in the context of TP53 alteration[27,28] cannot be eliminated.

**iHGGs recapitulate molecular and genetic hallmarks of GBM.** We further investigated if these iHGGs showed inter- and intra-tumor heterogeneity, which is another hallmark of GBM and cancer in general[29]. This important feature of cancer has not been well studied in previous models to date. To investigate the robustness of our iHGG models we performed in triplicate, single-cell RNA sequencing (scRNA-seq) using primary iHGG

**Fig. 2 Histology of iHGGs.** H&E staining of *PTEN*−/−;*NF1*−/− iHGGs showing a region of hypercellularity infiltrated by irregular, elongated to angulated tumor cells with occasional mitoses (**a**), scale bars, 50 μm (left) and 10 μm (right), biphasic dense (glial) and loose (mesenchymal/sarcoma) morphologies, typical of gliosarcoma (**b**), necrosis (central pink zone) with peripheral "pseudopalisading" of cells around the necrotic center (**c**), scale bars, 200 μm (**b**, **c**), vascular endothelial proliferation (**d**), scale bar, 100 μm, rupture through the pial surface, and consequently subarachnoid spread (upper right) (**e**), scale bar, 200 μm, and "secondary structures" typical of glioma, including perineuronal satellitosis and subpial accumulation of tumor cells (**f**), scale bar, 50 μm. GFAP (**g**), Olig2 (**h**), Ki-67 (**i**) staining of *PTEN*−/−;*NF1*−/− iHGGs, scale bars, 100 μm (**g**–**i**). H&E staining of *TP53*−/−;*PDGFRA*Δ8−9 iHGGs showing nodular growth of a primitive neuronal component (dark purple) intermingled with glial component (pink) (**j**), scale bar, 200 μm, rosettes with neuropil-like texture in a primitive neuronal component (**k**), scale bar, 50 μm, a serpiginous zone of pseudopalisading necrosis (**l**), scale bar, 200 μm, and a tumor rupture through ependyma illustrating intraventricular growth (**m**), scale bar, 500μm. GFAP (**n**), Olig2 (**o**), Ki-67 (**p**) staining of *TP53*−/−;*PDGFRA*Δ8−9 iHGGs, scale bars, 100 μm (**n**–**p**).

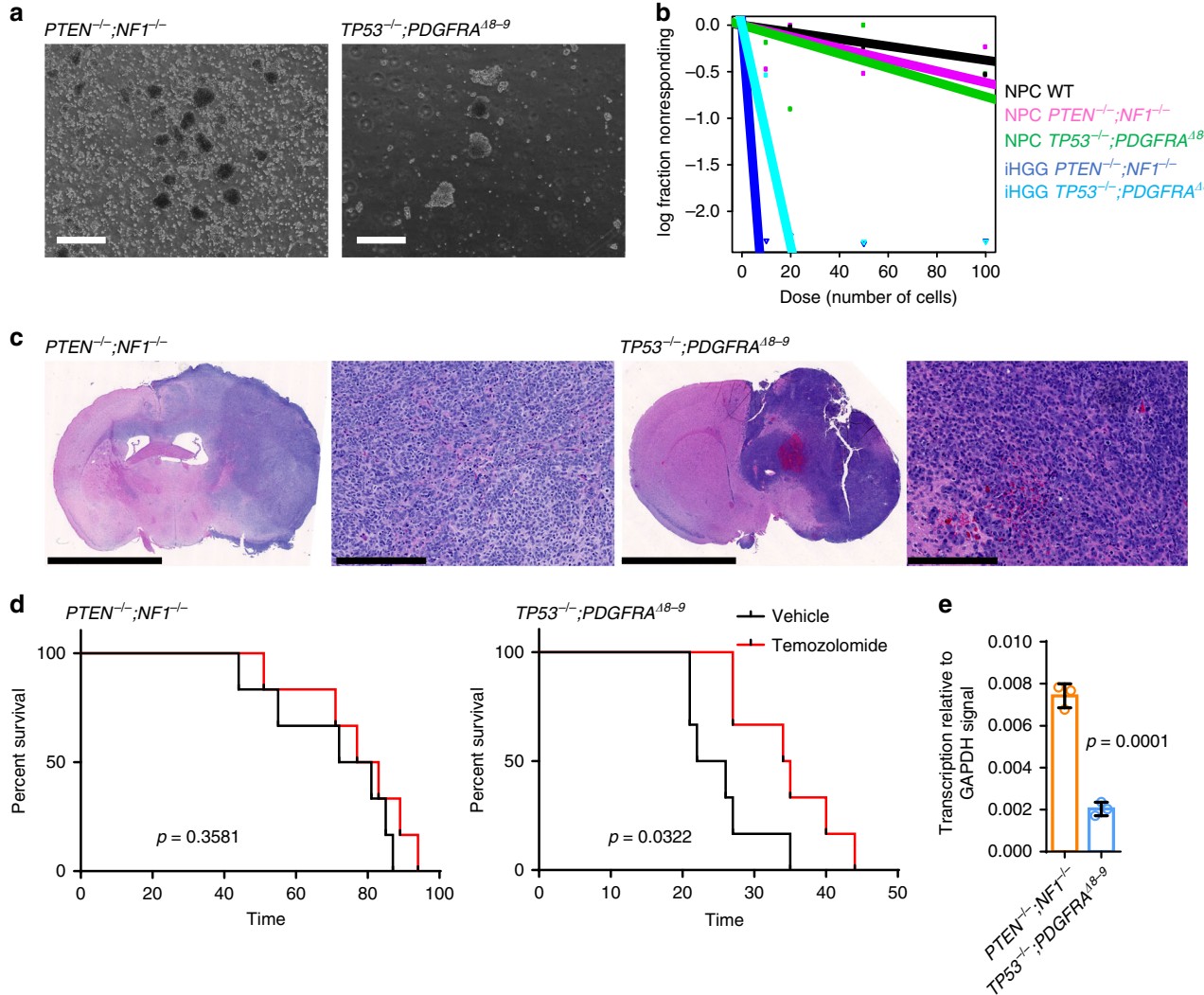

**Fig. 3 Cells from iHGG models can be cultured in vitro and re-engrafted to form secondary tumors with different drug response. a** iHGG spheres obtained by maintaining iHGG tumor cells in neurosphere culture conditions, scale bars, 2 mm. **b** Extreme limiting dilution analysis of input NPCs and tumor-derived iHGG sphere cells. **c** H&E staining of secondary tumors generated from re-engraftment of primary iHGG spheres, scale bars, 5 mm, 250 μm, 5 mm, 250 μm, (left to right). **d** In vivo survival assays of mice orthotopically engrafted with primary iHGG sphere cells upon treatment either with vehicle or temozolomide. Data are representative of six replicates, *n* = 6 animals for each treatment arm for each model. Data were analyzed by the log-rank test. **e** MGMT expression levels in iHGG cells analyzed by semi-quantitative RT-qPCR. Data are representative of three replicates, *n* = 3. Data are represented as mean ± SD, analyzed by unpaired *t*-test. Source data are provided as a Source Data file.

spheres, secondary iHGG tumor cells obtained from orthotopic injection of the primary spheres, as well as secondary spheres derived by in vitro culture of the secondary tumor cells for both genotypes, for a total of 14 samples (Fig. 4a).

Visual analysis of all the samples by Uniform Manifold Approximation and Projection (UMAP) reveals clear structural stratification between primary and secondary spheres of the same genotype as well as between spheres and tumors (Fig. 4b). However, the greatest variation appears between the two iHGG models of different genotypes. This inter-tumor heterogeneity between iHGG models was not apparent in pre-engraftment NPCs with different gene edits (Supplementary Fig. 8). In fact, the

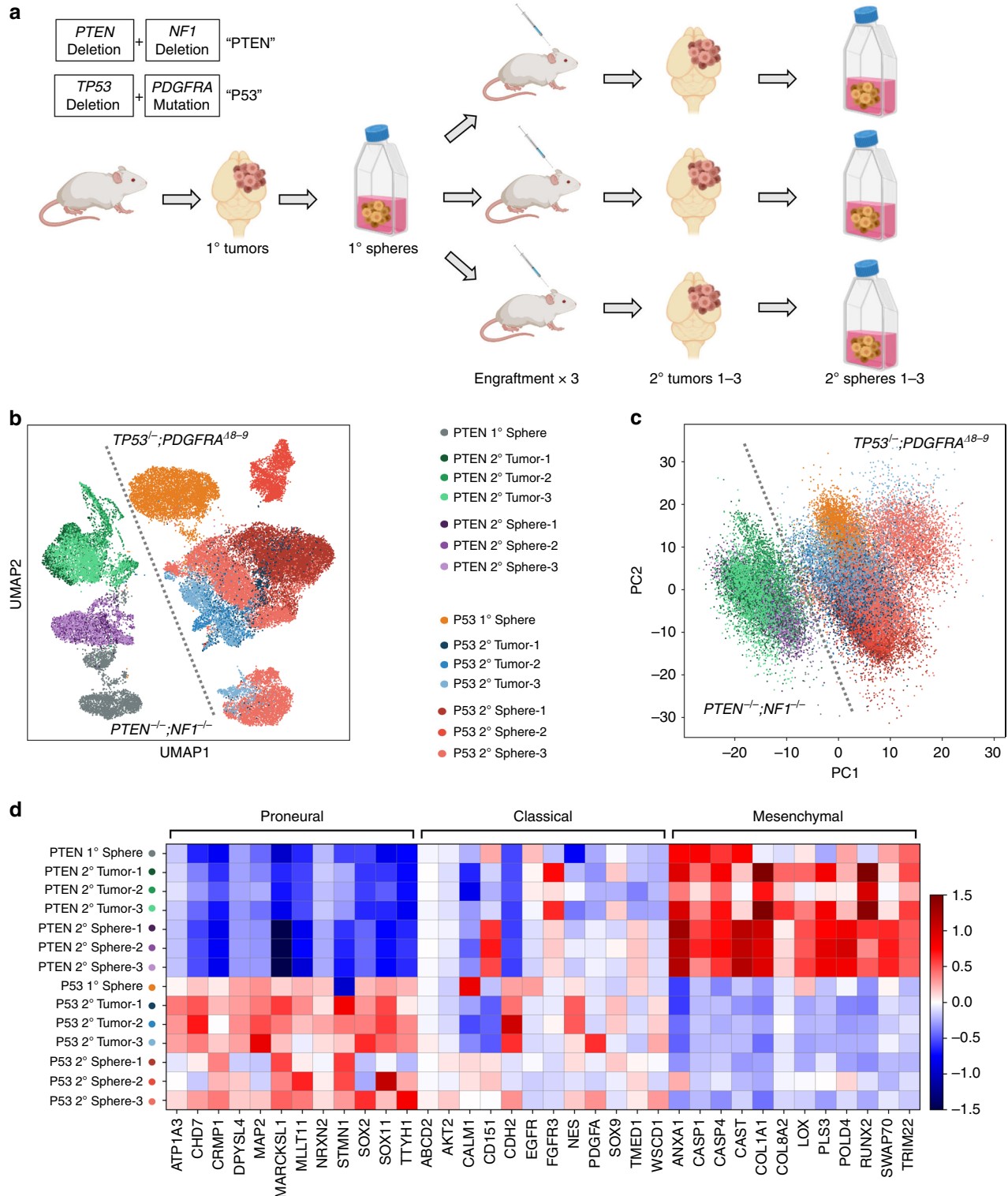

**Fig. 4 iHGG models present inter-tumor heterogeneities and divergent transcriptomes driven by molecular subtypes. a** Schema of scRNA-seq analysis of iGBMs. **b** Uniform Manifold Approximation and Projection (UMAP) analysis of all sequenced samples. **c** Principal component analysis of all sequenced samples (the color code is same as in **b**). **d** The heatmap of GBM molecular subtype analysis based on average gene expression of individual cells in each sample for a manually curated gene list based on ref. [18].

strongest driver of transcriptomic differences is genotype, as shown by the clear split between them, regardless of their origin (spheres or tumors, primary or secondary) for the strongest principal component (PC), pc1 (Fig. 4c). These findings continue to support the notion that a small number of driver mutations are sufficient for the development of such pathognomonic inter-tumor heterogeneity that arises through the process of transformation.

Given that our models were engineered to recapitulate different clinical GBM molecular subtypes, specifically proneural and

mesenchymal, we sought to determine if our samples manifested transcriptomic GBM signatures as established previously[8,18]. $TP53^{-/-};PDGFRA^{\Delta8-9}$ iHGG show upregulation of genes characteristic of the proneural subtype, while the $PTEN^{-/-};NF1^{-/-}$ iHGG show a mesenchymal subtype signature for both spheres and tumors (Fig. 4d, and Supplementary Fig. 9). Subtype scores involving all expressed genes under each subtype show similar trends, with higher proneural scores for $TP53^{-/-};PDGFRA^{\Delta8-9}$ iHGG and higher mesenchymal scores for $PTEN^{-/-};NF1^{-/-}$ iHGG (Supplementary Fig. 10). Interestingly, $TP53^{-/-};PDGFRA^{\Delta8-9}$ iHGG samples also show increased classical subtype scores. Importantly, when examined at single-cell resolution, each sample shows intra-tumor heterogeneity with different populations of cells presenting signatures of different subtypes (Supplementary Fig. 10), as is characteristic of GBM patient samples[29]. In addition, all samples are comprised of populations of cycling and noncycling cells (Supplementary Fig. 11a, b), which is also characteristic of patient samples and is in juxtaposition with other in vitro GBM models, where almost 100% of cells are cycling[29]. Finally, in agreement with previous literature on patient samples, cells with high proneural scores also score highly on stemness as is the case for our $TP53^{-/-};PDGFRA^{\Delta8-9}$ iHGG samples (Supplementary Fig. 11c–e)[29]. In conclusion, our results highlight the robustness of our iHGG models, for both spheres and tumors, in recapitulating hallmarks of patient GBM samples as is the case for cellular inter- and intra-tumor heterogeneity, subtype signatures and cycling and stemness scores.

**Distinct iHGGs present different patterns of tumor evolution.** It is also apparent from the UMAP plots that the transcriptomic signature of primary spheres evolves as they are passaged through mice, excised, and cultured in vitro. In fact, by analyzing separately the transition of each genotype model from primary to secondary spheres, we gain insights into the biology of the tumors as well as the role that in vivo passaging plays. We performed unsupervised Louvain clustering of $PTEN^{-/-};NF1^{-/-}$ iHGG primary and secondary spheres (Fig. 5a) and, in parallel, $TP53^{-/-};PDGFRA^{\Delta8-9}$ iHGG primary and secondary spheres (Fig. 5b) and found 8 and 15 distinct clusters, respectively. In both cases primary spheres are represented, almost exclusively, by unique clusters not found in any secondary spheres. Remarkably, all three $PTEN^{-/-};NF1^{-/-}$ iHGG secondary spheres are found in very similar proportions in the remaining clusters. In stark contrast, each of the $TP53^{-/-};PDGFRA^{\Delta8-9}$ iHGG secondary spheres show unique cluster makeups.

Moreover, when the differentially expressed genes of each cluster are subjected to gene ontology (GO) analysis, different patterns emerge for each iHGG model. $PTEN^{-/-};NF1^{-/-}$ iHGG spheres are subdivided in two broad categories, namely cell cycle or cell motility and extra- and intra-cellular fiber reorganization, with several cluster sharing a variety of GO terms (Fig. 5c). In fact, both categories appear to be exacerbated following tumor formation in mice as shown by the increase in the number of GO terms associated with each secondary sphere cluster. This increase in cell motility terms supports our observation regarding more prominent diffuse invasion of $PTEN^{-/-};NF1^{-/-}$ iHGG compared with $TP53^{-/-};PDGFRA^{\Delta8-9}$ iHGG (Supplementary Fig. 12). In

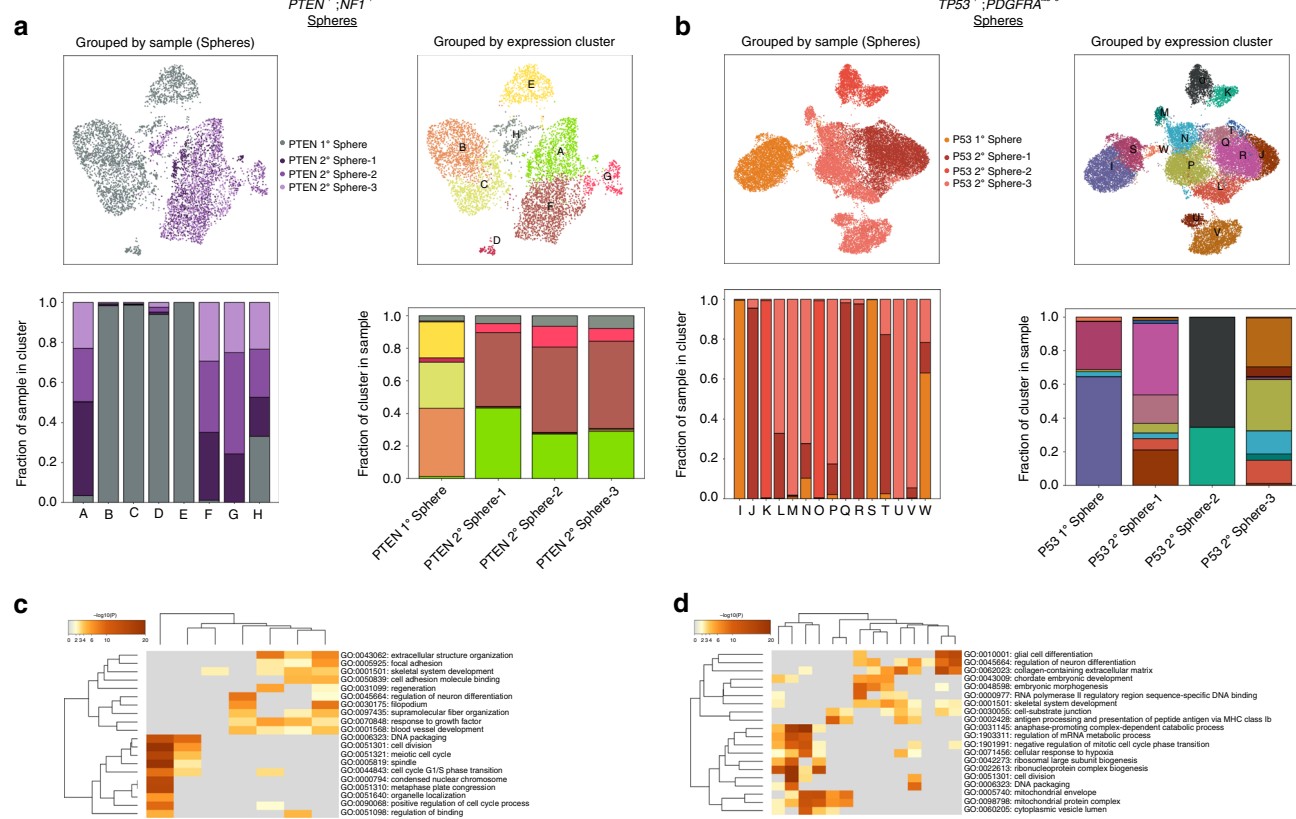

**Fig. 5 Genetically distinct iHGG models present different patterns of longitudinal evolution.** Primary and secondary spheres of $PTEN^{-/-};NF1^{-/-}$ (**a**) and $TP53^{-/-};PDGFRA^{\Delta8-9}$ (**b**). UMAP plot color-coded by samples (top left) or by Louvain clustering (top right). Sample distribution found in each Louvain cluster, color-coded by sample identity (bottom left) and Louvain cluster distribution per sample, color-coded by cluster identity (bottom right). Clustered heatmaps of enriched GO terms extracted from differentially expressed genes of each Louvain cluster in $PTEN^{-/-};NF1^{-/-}$ (**c**) and $TP53^{-/-};PDGFRA^{\Delta8-9}$ (**d**). Color scale represents statistical significance. Gray color indicates a lack of significance.

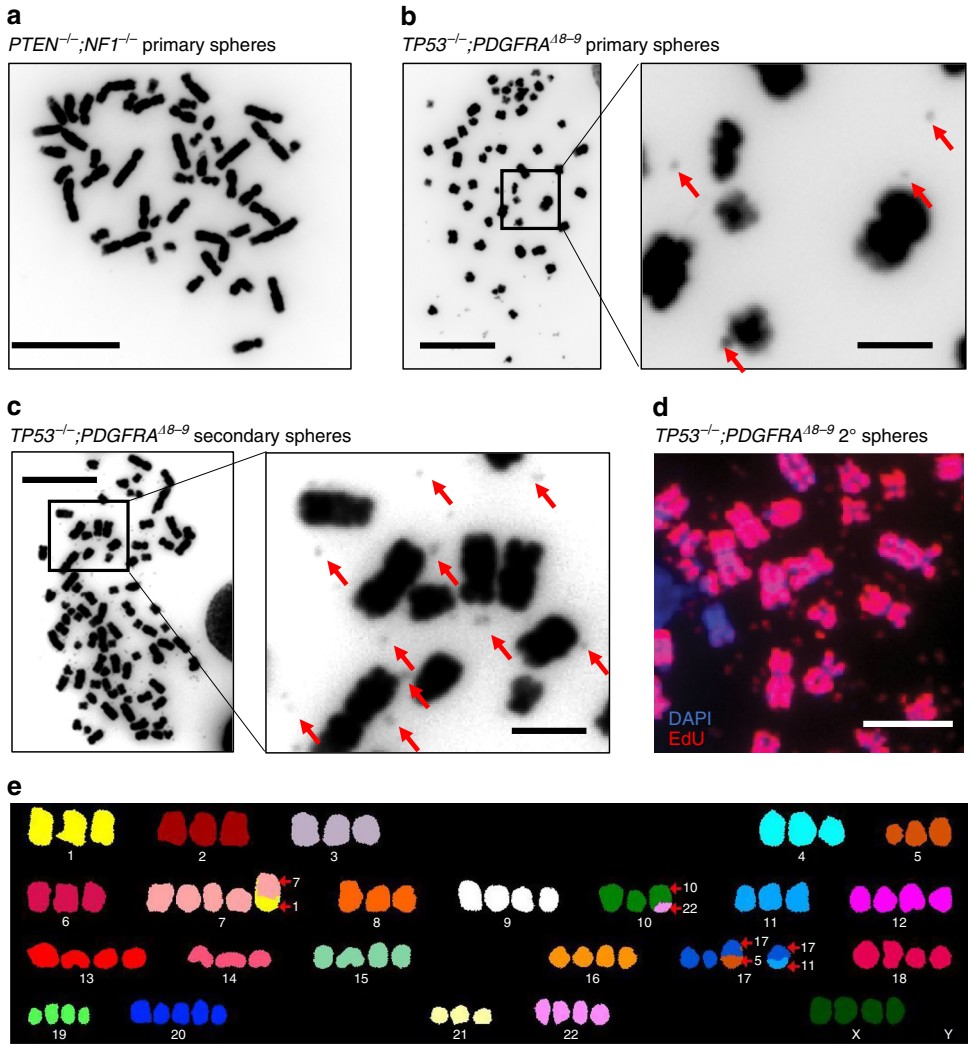

**Fig. 6 $TP53^{-/-};PDGFRA^{\Delta8-9}$ iHGG shows prominent karyotype abnormalities accompanied by extrachromosomal DNA. a** DAPI staining of $PTEN^{-/-};NF1^{-/-}$ primary iHGG cells, scale bar, 10 μm. **b** DAPI staining of $TP53^{-/-};PDGFRA^{\Delta8-9}$ primary iHGG cells. Red arrows indicate ecDNA, scale bars, 10 μm (left), 2 μm (right). **c** DAPI staining of $TP53^{-/-};PDGFRA^{\Delta8-9}$ secondary iHGG cells. Red arrows indicate ecDNA, scale bars, 10 μm (left), 2 μm (right). **d** EdU labeling of chromosomes and ecDNA in a metaphase spread of $TP53^{-/-};PDGFRA^{\Delta8-9}$ secondary iGBM, scale bar, 5 μm. **e** Spectral karyotyping analysis of $TP53^{-/-};PDGFRA^{\Delta8-9}$ iHGG cells.

contrast, in addition to cell cycle associated GO terms, $TP53^{-/-};PDGFRA^{\Delta8-9}$ iHGG spheres are represented by diverse GO terms, many of which are unique to each secondary sphere, including glial cell differentiation, response to hypoxia, embryonic morphogenesis and, similarly to $PTEN^{-/-};NF1^{-/-}$ iHGG spheres, cell motility associated terms (Fig. 5d). Likewise, the mesenchymal signature of $PTEN^{-/-};NF1^{-/-}$ iHGG spheres is homogenous across most clusters, whereas proneural scores are heterogeneous in nature (Supplementary Figs. 13 and 14). Overall, even though both iHGG models show transcriptional drift from primary to secondary spheres, $TP53^{-/-};PDGFRA^{\Delta8-9}$ iHGGs appear to show a less unidirectional path with increased heterogeneity.

We previously reported that extrachromosomal DNA (ecDNA) is prevalent in many cancer types, especially in GBM, and that ecDNA is associated with resistance to drug treatment and rapid evolution of tumor heterogeneity[30,31]. To determine if our iGBM models recapitulated the generation of ecDNA, we first investigated if the original input NPCs possessed karyotype abnormalities or traces of ecDNA. Based on DAPI staining of metaphase spreads and digital karyotyping, $PTEN^{-/-};NF1^{-/-}$ iHGG cells were karyotypically normal (Fig. 6a). In sharp contrast, metaphase

spreads of cells obtained from $TP53^{-/-};PDGFRA^{\Delta8-9}$ iHGGs showed small DAPI-stained dots adjacent to chromosomes, suggestive of ecDNA (Fig. 6b), consistent with our previous findings in GBM tumor samples[31]. Furthermore, double minute-like structures became more apparent in the secondary tumors obtained by re-engraftment of the primary spheres (Fig. 6c), and were replication competent as indicated by incorporation of EdU (Fig. 6d). The $TP53^{-/-};PDGFRA^{\Delta8-9}$ iHGGs also presented striking numerical and structural chromosome alterations (Fig. 6e). This supports a clonally unstable nature of the $TP53^{-/-};PDGFRA^{\Delta8-9}$ model, where genomic instability or ecDNA could be driving dynamic accelerated clonal evolution[31,32].

**iHGGs confirm features characteristic of patient samples**. We also applied unsupervised Louvain clustering of $PTEN^{-/-};NF1^{-/-}$ iHGG secondary tumors (Fig. 7a) and, in parallel, $TP53^{-/-};PDGFRA^{\Delta8-9}$ iHGG secondary tumors (Fig. 7b) to further compare inter- and intra-tumor variability of our iHGG tumor models and found 7 and 8 distinct clusters, respectively. Each one of the clusters is represented by a unique set of

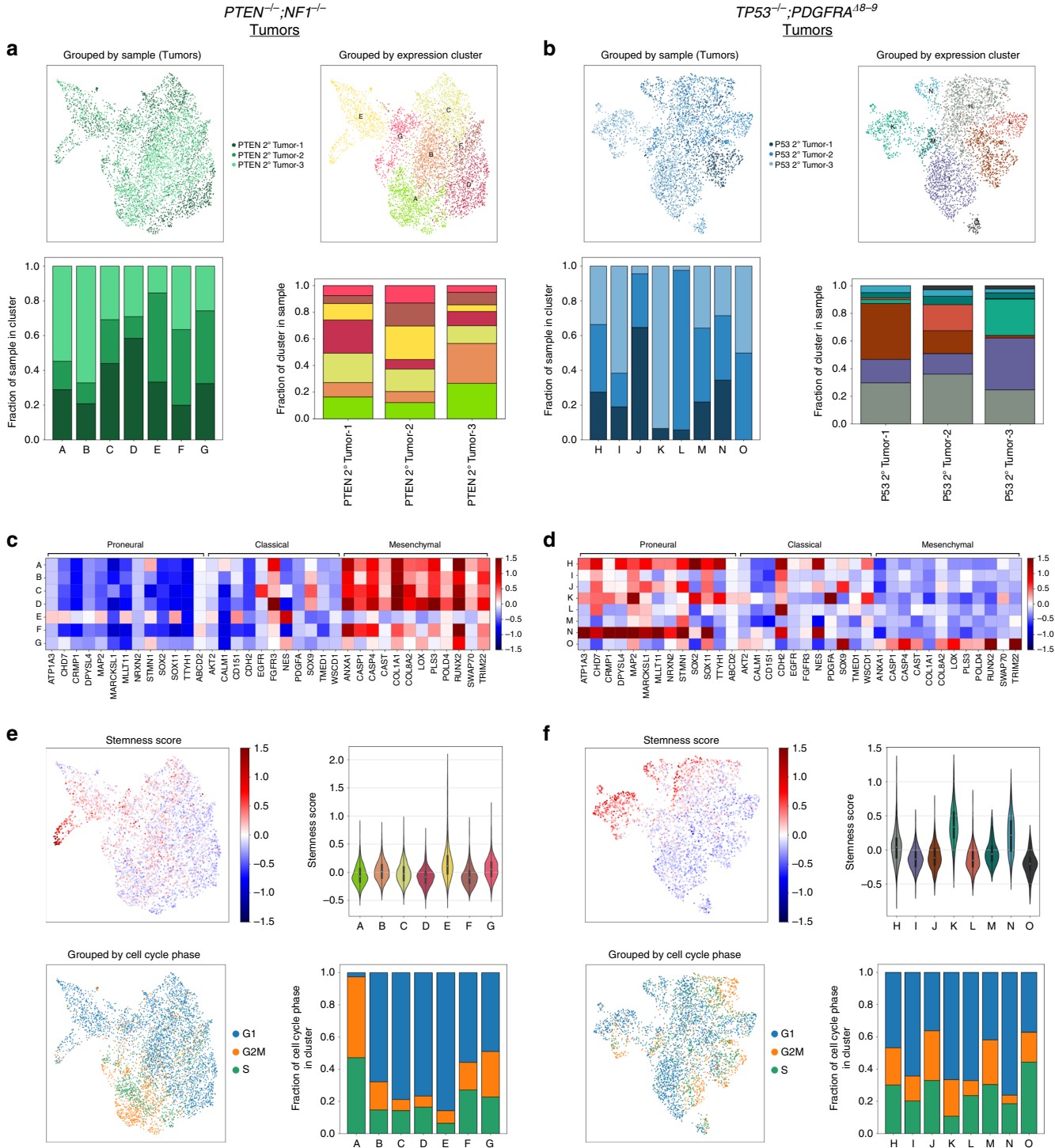

**Fig. 7 iHGG tumors confirm features characteristic of patient tumor samples.** $PTEN^{-/-};NF1^{-/-}$ (**a**) and $TP53^{-/-};PDGFRA^{\Delta8-9}$ (**b**) secondary tumors. UMAP plot color-coded by samples (top left) or by Louvain clustering (top right). Sample distribution found in each Louvain cluster, color-coded by sample identity (bottom left) and Louvain cluster distribution per sample, color-coded by cluster identity (bottom right). GBM molecular subtype analysis based on average gene expression of individual cells in each Louvain cluster for $PTEN^{-/-};NF1^{-/-}$ (**c**) and $TP53^{-/-};PDGFRA^{\Delta8-9}$ (**d**) tumors. For $PTEN^{-/-};NF1^{-/-}$ (**e**) and $TP53^{-/-};PDGFRA^{\Delta8-9}$ (**f**) tumors, stemness scores were calculated for each cell and results overlaid on a UMAP plot (top left) or summarized as violin plots for each cluster (top right). Cells were categorized based on their cell cycle status (G1, G2M or S), overlaid on a UMAP plot (bottom left) and their distribution was calculated for each Louvain cluster (bottom right).

differentially expressed genes which, if stratified by sample, show no clear patterns, as would be expected when intra-tumor heterogeneity is present at the single-cell level (Supplementary Fig. 15). Furthermore, $PTEN^{-/-};NF1^{-/-}$ iHGG secondary tumors appear more homogenous than their $TP53^{-/-};PDGFRA^{\Delta8-9}$

counterparts as each one of the triplicates is represented in all clusters. On the contrary, $TP53^{-/-};PDGFRA^{\Delta8-9}$ iHGG secondary tumors show clusters containing cells from all samples but also several clusters comprised of only two or almost exclusively one sample, indicating increased inter-tumor variability.

Significantly, when GBM molecular subtypes of each cluster are inspected, $PTEN^{-/-};NF1^{-/-}$ samples show, for the most part, a homogenous mesenchymal signature, with the exception of two clusters ('E' and 'G') (Fig. 7c). In stark contrast, subtype signatures of each cluster derived from $TP53^{-/-};PDGFRA^{\Delta8-9}$ tumors show unique combinations, with three clusters accounting for the majority of the proneural signature and smaller contributions by the remaining clusters with the exception of cluster 'O' which has a clear mesenchymal signature (Fig. 7d). Notably, for both models, intra-tumor heterogeneity is also observed as each cluster presents signatures of different molecular subtypes, a hallmark of GBM patient samples[29]. Finally, some $TP53^{-/-};PDGFRA^{\Delta8-9}$ tumor clusters also score higher for stemness in comparison to $PTEN^{-/-};NF1^{-/-}$ clusters; while both models show a heterogenous composition of cycling and noncycling cells (Fig. 7e, f). In conclusion, our isogenic models faithfully recapitulate HGG pathobiology, including inter- and intra-tumor heterogeneity, differential drug sensitivity, ecDNA amplifications, and rapid clonal evolution. Variations of this tumor avatar platform can be applied to different types of cancers and will allow, amongst other things, the study of clonal evolution, longitudinal assessment in vitro and in vivo, and genotype-based therapeutic vulnerabilities deciphered in an isogenic background.

## Discussion

We generated isogenic iHGG models from hiPSCs by introducing different combinations of genetic alterations characteristic of this disease. These models both recapitulated pathological features of HGG, but at the same time, displayed distinct mutation-dependent variation in histological morphology, gene expression, and ploidy. Some of our findings were consistent with previous mouse models. For example, NF1-deleted tumors and PDGF-driven tumors showed features of mesenchymal and proneural subtype, respectively in mouse models[33–35], where the former tumors were resistant to TMZ while the latter tumors were sensitive to the drug[33]. Such consistency among our models, patient samples, and previous models suggests the key roles of small numbers of driver mutations in determining tumor phenotypes. The heterogeneity presented in our models is an essential feature for the proper modeling of GBM as this prominent characteristic is a confounding aspect making these tumors difficult to treat. Although there have been several models generated from pluripotent cells and genome editing technologies to date[15], our iHGG models are distinct in the way that they present exact pathognomonic features of GBM, and reproduce the heterogeneity of the disease from isogenic cells. These approaches have been applied for models of other brain tumors such as medulloblastoma[36]. One limitation in our approach is that genome engineering was performed in hiPSCs, an irrelevant cell of origin for GBM. However, the fact that tumor models derived from appropriately differentiated NPCs from edited hiPSCs recapitulate GBM pathobiology suggests that our platform has potential for broader cancer modeling using various cell lineage differentiation protocols applied to hiPSCs. Furthermore, recent development of 3D in vitro brain tumor models using cerebral organoid suggests that several different combinations of oncogenic mutations give rise to expanding tumor-like components within organoids, which would approximate in vivo conditions better than conventional cell culture conditions[37]. Such applications of advanced differentiation technology of pluripotent stem cells and genome engineering enabling introduction of genetic alterations actually observed in patient samples, could further develop the next generation of cancer models.

A combination of $EGFR$ activation and inactivation of $Ink4a/Arf$, which are common co-occurring genetic alterations observed in high-grade gliomas, has been shown to play a role in dedifferentiation of astrocytes through the process of gliomagenesis[38]. How the genetic alterations we engineered in NPCs contribute to the formation of iHGGs is to be further studied, and presents an ideal platform to investigate mechanisms promoting transformation guided by cell lineage. Also, how the NPCs with GBM associated mutations, which prior to orthotopic engraftment do not show GBM subtype specific transcriptome signatures, present such signatures through the process of in vivo transformation, is to be further investigated.

Once xenograft tumors were obtained with our proneural and mesenchymal iHGG models, cells from these tumors maintained tumorigenicity and formed secondary tumors in vivo resembling the original tumors, as also seen in PDX models[39]. Owing to this characteristic, the iHGG cells can be passaged and maintained as cell lines once tumors are obtained. Further, as shown here, it is quite feasible to introduce different combinations of genetic alterations in hiPSCs, and such different edits result in divergent phenotypes. Thus, by expanding our gene-editing spectrum, we expect that these models would enable us to evaluate the influence of select driver genetic alterations found in different cancer types, which is less feasible with PDX models due to numerous acquired passenger mutations and genetic backgrounds that are highly variable sample to sample.

Another striking finding in our iHGG models was aneuploidy accompanied by ecDNA, observed in $TP53^{-/-};PDGFRA^{\Delta8-9}$ tumors. In our previous analyses, ecDNA was observed in more than 80% of PDX cell lines derived from GBM[31], suggesting that ecDNA formation is a fundamental feature in the pathogenesis of GBM. Interestingly, a mouse model with a combination of $CDKN2A^{-/-}$ and a PDGFRA point mutation showed brain tumors with double minute chromosomes or ecDNA[40], while another $CDKN2A^{-/-}$ mouse model that generated brain tumors after irradiation similarly had ecDNA[41]. Using human intestinal stem cells that were edited for the most commonly mutated colorectal cancer genes ($APC$, $TP53$, $KRAS$ and $SMAD4$), extensive aneuploidy occurred and these quadruple mutant cells grew as tumors in immunocompromised mice with features of invasive carcinoma[12]. Together with our results, these previous models suggest alterations in $TP53$ or $CDKN2A$, which are commonly affected in GBM as well as other cancer types, play an essential role in the genesis of chromosomal instability that results in aneuploidy or ecDNA formation. In summary, we propose a modeling system for HGG by introducing different combinations of essential genetic alterations in hiPSCs, which result in tumor avatars faithfully recapitulating histology, gene expression signatures, and cytogenetic features of HGG. As these avatars are faithfully expressing gene expression signatures characteristic of GBMs, we expect that these models will be a useful platform to study cancer biology based on genetic drivers, cell of origin defined by the differentiation program of genome-edited iPSCs, and possible other parameters such as xenograft location and gender differences.

## Methods

**Cell culture**. Experiments using human pluripotent stem cells were conducted under the regulations of the UCSD Human Research Protections Program, project number 151330ZX. Human iPS cells, CV-iPS-B cells were obtained from Dr Lawrence S. B. Goldstein[42]. CV-iPS-B cells were cultured on plates coated with Matrigel hESC-Qualified Matrix (Corning) in mTeSR1 media (Stemcell Technologies). NPCs were cultured on matrigel-coated plates in NPC maintenance media containing DMEM/F12 with GlutaMAX (Thermo Fisher Scientific), 1 × N-2 supplement (Thermo Fisher Scientific), 1 × B-27 supplement (Thermo Fisher Scientific), 50 mM ascorbic acid (Tocris), 3 μM CHIR99021 (Tocris) and 0.5 μM purmorphamine (Tocris). Sphere cells were cultured in suspension in DMEM/F12 with GlutaMAX (Thermo Fisher Scientific) with 1 × B-27 supplement, 20 ng/ml EGF (Stemcell Technologies) and 20 ng/ml bFGF (Stemcell Technologies).

**Generation of genetically engineered hiPSC clones.** A plasmid, pSpCas9(BB)-2A-GFP or px458, which expresses Cas9-T2A-GFP and sgRNA[43], was purchased from Addgene (Plasmid #48138). The designated sgRNA sequences for each of the targeted genes were cloned into px458 using combinations of top and bottom oligonucleotides listed below.

PTEN-intron 4-top: 5′-CACCGGAATTTACGCTATACGGAC-3′,
PTEN-intron 4-bottom: 5′-AAACGTCCGTATAGCGTAAATTCC-3′,
PTEN-intron 5-top: 5′-CACCGAACAAGATCTGAAGCTCTAC-3′,
PTEN-intron 5-bottom: 5′-AAACGTAGAGCTTCAGATCTTGTTC-3′,
TP53-intron 1-top: 5′-CACCGGGTTGGAAGTGTCTCATGC-3′,
TP53-intron 1-bottom: 5′-AAACGCATGAGACACTTCCAACCC-3′,
TP53-intron 6-top: 5′-CACCGCATCTCATGGGGGTTATAGGG-3′,
TP53-intron 6-bottom: 5′-AAACCCCTATAACCCCATGAGATGC-3′,
NF1-intron 31-top: 5′-CACCGGATAGCACTCTTCCCGAGCTA-3′,
NF1-intron 31-bottom: 5′-AAACTAGCTCGGGAAGAGTGCTATC-3′,
NF1-intron 33-top: 5′-CACCGCCTTTGGGGAGGTCTTTCGTC-3′,
NF1-intron 33-bottom: 5′-AAACGACGAAAGACCTCCCCAAAGC-3′,
PDGFRA-intron 7-top: 5′-CACCGATTTGTATGTAGCGGTCTGC-3′,
PDGFRA-intron 7-bottom: 5′-AAACGCAGACCGCTACATACAAATC-3′,
PDGFRA-intron 9-top: 5′-CACCGCCACGGGAACACTCTAAGA-3′,
PDGFRA-intron 9-bottom: 5′-AAACTCTTAGAGTGTTCCCGTGGC-3′.

Each of top and bottom oligonucleotides were phosphorylated and annealed by incubating 10 μM each of oligonucleotides, 1 × T4 DNA ligase buffer (New England Biolabs), 5U T4 polynucleotide kinase (New England Biolabs) at 37 °C for 30 min, 95 °C for 5 min and by cooling down to 25 °C at 0.1 °C/s using a thermocycler. Annealed oligonucleotides were cloned into px458 by incubating 25 ng px458, 1 μM annealed oligonucleotides, 1× CutSmart buffer (New England Biolabs), 1 mM ATP (New England Biolabs), 10U BBSI-HF (New England Biolabs) and 200U T4 ligase (New England Biolabs) at 37 °C for 5 minutes, 23 °C for 5 min for 30 cycles. Correct cloning of each sgRNA sequence was confirmed by Sanger sequencing using U6 sequencing primer: 5′-GATACAAGGCTGTTAGAGAGATAATT-3′.

Human iPSCs were cultured in 10 μM Y-27632 RHO/ROCK pathway inhibitor for 2 h before dissociation. The cells were dissociated to single cells using Accutase (Innovative Cell Technologies). The dissociated hiPSCs ($1 × 10^6$ cells) were resuspended in 100 μl of supplemented solution of the Human Stem Cell Nucleofector Kit 1 (Lonza) containing 8 μg total of a combination of px458 plasmids targeting each gene and electroporated using B-016 program of Nucleofector 2b (Lonza). Electroporated hiPSCs were cultured on matrigel-coated plates in mTeSR1 for 48 h. GFP-positive cells were then sorted by flow cytometer (SH800, SONY) and $1–2 × 10^4$ sorted cells were plated on a 10-cm matrigel-coated plate in mTeSR1. Isolated colonies were manually picked and plated in duplicated matrigel-coated 96-well plates.

The hiPSCs clones on one of the duplicated 96-well plates were lysed using QuickExtract DNA Extraction Solution (Epicenter) and genotyping PCR was performed using Platinum Taq DNA Polymerase High Fidelity (Thermo Fisher Scientific) in 10-μl reaction volume containing 0.2 μM of each primer with the following reaction conditions: 94 °C for 2 min, 40 cycles of 94 °C for 15 s, 55 °C for 30 s, and 68 °C for 4 min. The PCR amplicons were visualized in agarose gels (Supplementary Fig. 1). Primers used for the genotyping PCR are listed below.

PTEN-i4-f: 5′-GAGTCCTGACGAAATGTCCATG-3′,
PTEN-i5-r: 5′-CCTGTT TTCCAGGGACTGAG-3′,
NF1-i31-f: 5′-ACTCTGGAAAGGGATGGGAG-3′,
NF1-i33-r: 5′-CCGGCTTCAGCTTCAAAGTAG-3′,
TP53-i1-f: 5′-CCGATCACCTGAAGTAAGGAG-3′,
TP53-i6-r: 5′-CCTTAGCCTCTGTAAGCTTCAG-3′,
PDGFRA-i7-f: 5′-TGTACTCCTGTCCCCAGCTG-3′,
PDGFRA-i9-r: 5′-TCCTGAGAGTCATGGCAATG-3′.

Total RNA was extracted from the edited hiPSCs using RNeasy Plus Mini Kit (Qiagen) and was reverse transcribed using RNA to cDNA EcoDry Premix (Clontech) according to the manufacturer's instruction. Triplicate qPCR reactions containing cDNA obtained from 10 ng equivalent RNA were run on a CFX96 Real Time System (Bio-Rad) to confirm designated targeting of the genes with the following reaction conditions: 95 °C for 5 min, 40 cycles of 95 °C for 15 s, 56 °C for 30 s. Primer pairs were designed to span the deleted regions of each target gene. The data were normalized to GAPDH and the relative transcript levels were determined using 2-ΔCt formula. Primers used for the RT-qPCR are listed below.

GAPDH-RT-f: 5′-AATTTGGCTACAGCAACAGGGTGG-3′,
GAPDH-RT-r: 5′-TTGATGGTACATGACAAGGTGCGG-3′,
PTEN-RT-f: 5′-CGAACTGGTGTAATGATATGT-3′,
PTEN-RT-r: 5′-CATGAACTTGTCTTCCCGT-3′,
NF1-RT-f: 5′-GCCACCACCTAGAATCGAAAG-3′,
NF1-RT-r: 5′-AGCAAGCACATTGCCGTCAC-3′,
TP53-RT-f: 5′-CCAAGTCTGTGACTTGCACG-3′,
TP53-RT-r: 5′-GTGGAATCAACCCACAGCTG-3′,
PDGFRAΔ8–9-RT-f: 5′-GATGTGGAAAAGATTCAGGAAATAAGATG-3′,
PDGFRAwt-RT-f: 5′-CGCCGCTTCCTGATATTGAG-3′,
PDGFRA-RT-r: 5′-CTCCACGGTACTCCTGTCTC-3′.
The qPCR products were visualized by agarose gel electrophoresis.

**Differentiation of hiPSCs to neural progenitor cells.** Generation of small molecule neural progenitor cells (smNPCs) from iPSCs was adapted from a

previous study 21. In detail, human iPSCs at 70–80% confluency were dissociated using accutase (Innovative Cell Technologies) and resuspended at $1 × 10^6$ cells/ml in N2B27 medium (DMEM/F12 with GlutaMAX (Thermo Fisher Scientific), 1 × N-2 supplement (Thermo Fisher Scientific), 1 × B-27 supplement (Thermo Fisher Scientific), 150 mM ascorbic acid (Tocris), and 1% Penicillin/Streptomycin) supplemented with 1 μM Dorsomorphin (Tocris), 10 μM SB431542 (Tocris), 3 μM CHIR99021, 0.5 μM Purmorphamine and 5 mM Y-26732 (Stemcell Technologies). Three million cells were transferred into one well of an uncoated six-well tissue culture plate and incubated at 37 °C, 5% $CO_2$ on a shaker at 90 rpm. Uniform small EBs formed within 24 h and increased in size over the following days. After 48 h, a full media change was performed with N2B27 medium supplemented with Dorsomorphin, SB431542, CHIR99021, and Purmorphamine. At this time, about 2/3 of EBs were either discarded or EBs were split across three wells of a six-well plate to reduce the high cell density required initially to ensure uniform formation of embryoid bodies. On days 3–5, half media change was performed with fresh N2B27 media supplemented with Dorsomorphin, SB431542, CHIR99021, and Purmorphamine. On day 6, Dorsomorphin and SB431542 were withdrawn and a full media change with smNPC media (N2B27 media supplemented with 3 μM CHIR99021 and 0.5 μM Purmorphamine) was performed. At this stage, neuroepithelial folds were clearly visible in all EBs. On day 8, EBs were triturated by pipetting 10–15 times with a P1000 pipette and plated onto matrigel-coated 10 cm plates. After 3–4 days, attached EB fragments and outgrown cells were dissociated to single cells with accutase (Innovative Cell Technologies) and split at a 1:6–1:8 ratio onto matrigel-coated plates. After the first passage, cells were passaged at a 1:10–1:15 ratio every 3–6 days. For the first few passages, large flat non-smNPCs could be observed between smNPC colonies, but progressively disappeared no later than passages 3–6 in almost all cell lines. Total RNA was extracted from the differentiated smNPCs using the RNeasy Plus Mini Kit and was reverse transcribed using RNA to cDNA EcoDry Premix according to the manufacturer's instruction. Triplicate qPCR reactions containing cDNA obtained from 10 ng equivalent RNA were run on a CFX96 Real Time System to confirm NPC differentiation with the following reaction conditions: 95 °C for 5 min, 40 cycles of 95 °C for 15 s, 56 °C for 30 s. The data were normalized to GAPDH and the relative transcript levels were determined using 2-ΔCt formula. Primers used for the RT-qPCR are listed below.

Nanog-RT-f: 5′-GAAATACCTCAGCCTCCAGC-3′,
Nanog-RT-r: 5′-GCGTCACACCATTGCTATTC-3′,
Oct4-RT-f: 5′-AGAACATGTGTAAGCTGCGG-3′,
Oct4-RT-r: 5′-GTTGCCTCTCACTCGGTTC-3′,
Nestin-RT-f: 5′-GGTCTCTTTTCTCTTCCGTCC-3′,
Nestin-RT-r: 5′-CTCCCACATCTGAAACGACTC-3′,
Pax6-RT-f: 5′-GCCCTCACAAACACCTACAG-3′,
Pax6-RT-r: 5′-TCATAACTCCGCCCATTCAC-3′,
Sox1-RT-f: 5′-CAGCAGTGTCGCTCCAATCA-3′,
Sox1-RT-r: 5′-GCCAAGCACCGAATTCACAG-3′.
Spontaneous differentiation of NPCs was performed by maintaining NPCs on matrigel-coated plates in DMEM supplemented with 10% FBS for a week.

**Intracranial tumor formation.** Animal research experiments were approved by the UCSD Animal Care Program, protocol number S00192M, and were performed under its regulations. Wildtype and edited smNPCs were dissociated using accutase (Innovative Cell Technologies), washed with PBS, and resuspended at $1 × 10^6$ cells in 2 μL PBS supplemented with 0.1% BSA per animal. Resuspended cells were kept on ice and were inoculated into the striatum of 4–6 week-old female Nod scid mice (Charles River Laboratory) by stereotactic injections (1.0 mm anterior and 2.0 mm right to the bregma, and 3 mm deep from the inner plate of the skull). Wildtype hiPSCs were injected as a control as well.

**Immunohistochemistry.** Paraffin-embedded tissue blocks were sectioned using the UCSD Moore's Cancer Center Pathology Core and the Center for Advanced Laboratory Medicine (UCSD). Tissue sections were stained with antibodies to GFAP (1:6000) from Dako (Cat #Z0334), Olig2 (1:300) from EMD Millipore (Cat #Ab6910; Lot #3018858) Ki-67 (pre-diluted47) from Ventana Medical Systems (Cat #760-4286) and NM95 (1:300) from Abcam (Cat # ab190710). Slides were stained on a Ventana Discovery Ultra. Antigen retrieval was performed using CC1 for 24–40 min at 95 °C (or protease 2 for 12 min for GFAP). The primary antibodies were incubated on the sections for 32 min at 37 °C. Primary antibodies were visualized using the OmniMap system (HRP-labeled goat anti-mouse or rabbit; Ventana Medical systems) and used DAB as a chromagen followed by hematoxylin as a counterstain. Slides were rinsed, dehydrated through alcohol and xylene and coverslipped immunohistochemical stains for GFAP and Olig2 were graded as follows: 0 (no cells stained), 1+ (<5% tumor cells stained), 2+ (5–50% cells stained), and 3+ (>50% cells stained). Ki-67 positivity was manually counted on six different high power fields for each model.

**Sphere cell culture of iHGG.** Tumors were excised from mouse brains and cut in small pieces using a scalpel, and then incubated in 3.6 ml of Hank's Balanced Salt Solution (HBSS, Sigma) with 0.4 ml of 10× Trypsin solution (Sigma) at 37 °C for 20 min. After incubation, 200 μl of 10 mg/ml DNaseI stock solution (Sigma) was added and incubated for 60 s, and then 6 ml of HBSS was added to neutralize

Trypsin and DNaseI. Tumor tissue was resuspended by pipetting up and down several times through a glass Pasteur pipette. Dissociated tissue was filtered through a strainer and was spun down by centrifugation at $400 \times g$ for 3 min. Cells were resuspended in 1 ml of PBS and 9 ml of ACK lysing buffer (Invitrogen) and were incubated at 37 °C for 10 min to remove red blood cells. Approximately $1 \times 10^6$ cells were resuspended in 100 μl of MACS/BSA buffer (Miltenyi Biotec) and were incubated with 2 μl of Fc blocking solution (BioLegend) for 5 min on ice. After blocking, 5 μl of PE-conjugated antihuman HLA-A,B,C antibody (BioLegend) was added and cells were incubated for 15 min on ice. Stained cells were washed twice with 500 μl of MACS/BSA buffer. PE-positive cells were then sorted using a flow cytometer (SH800, SONY). Sorted human iHGG cells were maintained in DMEM/F12 with GlutaMAX (Thermo Fisher Scientific) with $1 \times$ B-27 supplement (Thermo Fisher Scientific), 20 ng/ml EGF (Stemcell Technologies) and 20 ng/ml bFGF (Stemcell Technologies).

**Extreme limiting dilution assay**. Extreme limiting dilution assay was performed based on a previous literature[24]. In detail, NPCs and iHGG spheres were dissociated into single cells using accutase (Innovative Cell Technologies) and 1, 5, 10, 20, 50, and 100 cells/well were plated in 96-well plates with five replicates for each experimental condition. The total number of spheres, per well and per treatment, were quantified after 14 days in culture. Data were analyzed by extreme limiting dilution analysis (http://bioinf.wehi.edu.au/software/elda/).

**Secondary tumor models and temozolomide treatment**. Primary iHGG spheres were dissociated using accutase (Innovative Cell Technologies), washed with PBS, and resuspended at $2.5 \times 10^5$ cells in 2 μl PBS supplemented with 0.1% BSA per animal. Resuspended cells were kept on ice and were inoculated as indicated above. Treatment of the mice started 7 days after inoculation of the iHGG cells by intraperitoneal injection of either vehicle (DMSO) or 50 mg/kg of TMZ (Selleckchem). The mice were treated once daily for the first 3 days followed by 2-day drug holidays, and then once daily for 2 days again followed by 2-day drug holidays and another set of 2-day once daily treatment. This set of treatment was repeated every 4 weeks and percentage of surviving mice over time was recorded. RT-qPCR to evaluate MGMT expression was run on a CFX96 real time system to confirm differentiation to NPCs with the following reaction conditions: 95 °C for 5 min, 40 cycles of 95 °C for 15 s, 56 °C for 30 s. The data were normalized to GAPDH and the relative transcript levels were determined using 2-ΔCt formula. Primers used for the RT-qPCR are listed below.

MGMT-RT-f: 5′-GCTGAATGCCTATTTCCACCA-3′,
MGMT-RT-r: 5′-CACAACCTTCAGCAGCTTCCA-3′,

**Cytogenetics**. Metaphase cells were obtained by treating cells with Karyomax (Gibco) at a final concentration of 0.1 μg/ml for 1–3 h. Cells were collected, washed in PBS, and resuspended in 0.075 M KCl for 15–30 min. Carnoys fixative (3:1 methanol/glacial acetic acid) was added dropwise to stop the reaction. Cells were washed an additional three times with Carnoys fixative, before being dropped onto humidified glass slides for metaphase cell preparations. DAPI was added to the slides. Images were captured with an Olympus FV1000 confocal microscope.

Spectral karyotyping analysis was performed at Applied Spectral Imaging.

Genomic DNA extracted from NPCs and iHGG cells using DNeasy blood and tissue kit (Qiagen) was analyzed by digital karyotyping using Illumina HiScan system (Illumina).

To detect DNA replication, cells were labeled with EdU and detected using the Click-iT Plus EdU Alexa Fluor 594 imaging kit (Invitrogen). Briefly, cells were pulse labeled with EdU (10 μM) for 1 h, then allowed to progress to metaphase for 12 h. KaryoMax (0.1 μg/ml) was added for 3 h to arrest cells in metaphase. The cells were then collected and metaphase spreads were prepared[31]. Cells in metaphase were dropped onto a glass slide, and EdU was detected by applying the Click-iT reaction cocktail directly onto the slides for 20 min at room temperature. Slides were then washed with 2× SSC and mounted with antifade mounting medium containing DAPI. Cells in metaphase were imaged using an Olympus BX43 fluorescent microscope equipped with a QIClick camera.

**RNA sequencing**. Total RNA was assessed for quality using an Agilent Tapestation, and all samples had RNA Integrity Numbers above 9.0. RNA libraries were generated using llumina's TruSeq Stranded mRNA Sample Prep Kit (Illumina) following manufacturer's instructions. RNA-seq reads were aligned to the human genome (hg19) with STAR 2.4.0 h (outFilterMultimapNmax 20, outFilterMismatchNmax 999, outFilterMismatchNoverLmax 0.04, outFilterIntronMotifs RemoveNoncanonicalUnannotated, outSJfilterOverhangMin 6 6 6 6, seedSearchStartLmax 20, alignSJDBoverhangMin 1) using a gene database constructed from Gencode v19[44,45]. Reads that overlap with exon coordinates were counted using HTSeqcount (-s reverse -a 0 -t exon -i gene_id -m union)[46,47]. Raw read counts were processed with DESeq2[48] and only genes with mean read count over 20 were considered for the analysis. Raw read counts were transformed using the variance stabilizing transformation function included in DESeq2[49]. Mean and standard deviation of normalized expression were calculated for each gene and Z-scores were determined by subtracting the mean from each expression value and dividing by the standard deviation.

**scRNA-seq and analysis**. For the scRNA-seq of secondary tumor cells, the tumors were dissected from mouse brains, cut into small pieces, and then incubated in HBSS (Sigma) containing 1× trypsin (Sigma) at 37 °C for 20 min, followed by mechanical dissociation using glass pipettes to obtain single cells. Cultured sphere cells were dissociated using accutase (Innovative Cell Technologies). Single cells were processed through the Chromium Single-Cell Gene Expression Solution using the Chromium Single Cell 3′ Gel Bead, Chip and Library Kits v2 (10× Genomics) as per the manufacturer's protocol. In brief, single cells were resuspended in 0.04% BSA in PBS. Ten thousand total cells were added to each channel with an average recovery of 3040 cells. The cells were then partitioned into Gel Beads in Emulsion in the Chromium instrument, where cell lysis and barcoded reverse transcription of RNA occurred, followed by amplification, shearing and 5′ adapter and sample index attachment. Agilent High Sensitivity D5000 ScreenTape Assay (Aglient Technologies) was performed for QC of the libraries. Libraries were sequenced on an Illumina NovaSeq. De-multiplexing, alignment to the hg19 transcriptome and unique molecular identifier-collapsing were performed using the Cellranger toolkit (version 2.0.1) provided by 10× Genomics. A total of 42,558 cells with ~53,000 mapped reads per cell were processed. Analysis of output digital gene expression matrices was performed using the Scanpy v1.3.3 package[50]. Matrices for all samples were concatenated and all genes that were not detected in at least 20 single cells were discarded, leaving 20,521 genes for further analyses. Cells with fewer than 600 or more than 8000 expressed genes as well as cells with more than 80,000 transcripts or 0.1% mitochondrial expressed genes were removed from the analysis. For the different sample subset combination analysis filtering steps were the same with the exception of specific gene and transcript thresholds for which cells were removed: $PTEN^{-/-};NF1^{-/-}$ spheres (fewer than 600 or more than 7000 expressed genes and more than 50,000 transcripts), $PTEN^{-/-};NF1^{-/-}$ tumors (fewer than 600 or more than 8000 expressed genes and more than 80,000 transcripts), $TP53^{-/-};PDGFRA^{\Delta8-9}$ spheres (fewer than 600 or more than 7000 expressed genes and more than 70,000 transcripts), $TP53^{-/-};PDGFRA^{\Delta8-9}$ tumors (fewer than 600 or more than 6000 expressed genes and more than 40,000). Data were log normalized and scaled to 10,000 transcripts per cell. Top 4000 variable genes were identified with the filter_genes_dispersion function, flavor = 'cell_ranger'. PCA was carried out, and the top 25 principal components were retained (21, 20, 23, and 24 for $PTEN^{-/-};NF1^{-/-}$ spheres, $PTEN^{-/-};NF1^{-/-}$ tumors, $TP53^{-/-};PDGFRA^{\Delta8-9}$ spheres and $TP53^{-/-};PDGFRA^{\Delta8-9}$ tumors, respectively). With these principal components, neighborhood graphs were computed with 20 neighbors and standard parameters with the pp.neighbors function. Louvain clusters were computed with the tl.louvain function and standard parameters (and 0.4, 0.4, 0.7 and 0.4 resolution for $PTEN^{-/-};NF1^{-/-}$ spheres, $PTEN^{-/-};NF1^{-/-}$ tumors, $TP53^{-/-};PDGFRA^{\Delta8-9}$ spheres and $TP53^{-/-};PDGFRA^{\Delta8-9}$ tumors, respectively). Single cell and mean expression per sample heatmaps were generated with the pl.heatmap and pl.matrixplot functions, respectively. Single-cell scores for TCGA molecular subtypes as well as stemness and cell cycle genes (see Supplementary Data 1) were computed with the tl.score_genes and tl.score_genes_cell_cycle functions, respectively. Differentially expressed genes were determined for each set of Louvain clusters with the tl.rank_gene_groups function (method = 'wilcoxon'). For GO analysis of primary and secondary spheres, differentially expressed genes of each Louvain cluster with log2fold over 0.5 and p-adjusted values under 0.05 were used as inputs (see Supplementary Data 2) on Metascape [51] (multiple gene list option and standard parameters), using all expressed genes in the 14 samples as background. Enrichment results are summarized in Supplementary Data 3.

**Statistical analyses**. All statistical analyses were performed using GraphPad Prism 6 software. Data are representative of results obtained in at least three independent experiments. Data sets were analyzed by unpaired $t$-test to determine significance ($p < 0.05$). Kaplan–Meier curves and comparison of survival were analyzed using Log-rank (Mantel–Cox) test.

**Reporting summary**. Further information on research design is available in the Nature Research Reporting Summary linked to this article.

## Data availability

Single-cell RNA sequencing and bulk RNA sequencing data have been deposited in the Gene Expression Omnibus database under GSE133479 [https://www.ncbi.nlm.nih.gov/geo/query/acc.cgi?acc = GSE133479] and GSE133509 [https://www.ncbi.nlm.nih.gov/geo/query/acc.cgi?acc = GSE133509] respectively. All the other data supporting the findings of this study are available within the article and its supplementary information files and from the corresponding author upon reasonable request. A reporting summary for this article is available as a Supplementary Information file.

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

## Acknowledgements

We thank W.K. Cavenee, A.H. Thorne, C. Zanca, Yeo lab and Furnari lab members for discussions and helpful suggestions. We are grateful to the IGM Genomics Center, University of California San Diego for conducting RNA sequencing, digital karyotyping, single-cell RNA sequencing and whole genome sequencing. We thank D. Pizzo at the Center for Advanced Laboratory Medicine, University of California San Diego for assistance with immunohistochemistry. This work was supported by National Institutes of Health (NIH) R01NS080939 (F.B.F.), R01HL137223 and R01HD85902 (G.W.Y.), R01GM114362 (V.B. and N.D.N.), P30CA023100 (IGM Genomics Center), the Defeat GBM Research Collaborative, a subsidiary of the National Brain Tumor Society (F.B.F. and P.S.M.), Ruth L. Kirschstein Institutional National Research Award T32 GM008666 (A.D.P.), grants from National Institute of Neurological Disorders and Stroke NS73831 (P.S.M.), the Ben and Catherine Ivy Foundation (P.S.M.), and National Science Foundation DBI-1458557 (V.B. and N.D.N.). I.A.C. is a San Diego IRACDA Fellow supported by NIH/NIGMS K12 GM068524 Award.

## Author contributions

T.K. designed the study, conducted the experiments and wrote the paper with inputs from other authors. I.A.C. performed bioinformatics analyses and wrote the paper. J.A.B., A.D.P., K.M.T., F.M.H., J.M., Sh.M. and A.D.B. conducted the experiments. Se.M. designed the study. R.F.H. and S.S. conducted pathological analyses. M.D. and N.D.N. performed bioinformatics analyses. J.R., K.A.F., V.B., C.C.C. and P.S.M. supervised parts of the study. G.W.Y. and F.B.F. conceived the study, wrote the paper with inputs from

other authors, and supervised all aspects of the study. T.K. and I.A.C. contributed equally to this work. J.A.B. and Se.M. contributed equally to this work.

## Competing interests

The authors declare the following competing interests: P.S.M. is a co-founder of Boundless Bio, Inc. (BB). He has equity interest in the company and serves as the chair of the Scientific Advisory Board. V.B. is a co-founder, serves on the scientific advisory board and has an equity interest in BB and Digital Proteomics, LLC (DP), and receives income from DP. The terms of this arrangement have been reviewed and approved by the University of California, San Diego in accordance with its conflict of interest policies. BB and DP were not involved in the research presented here. K.M.T became an employer of Boundless Bio after submission of this manuscript. G.W.Y. is a co-founder, member of the Board of Directors, equity holder, and paid consultant for Eclipse BioInnovations. The terms of these arrangements have been reviewed and approved by the University of California, San Diego in accordance with its conflict of interest policies. The remaining authors declare no competing interests.
