## [Peer Review File · Nature Communications]

REVIEWERS' COMMENTS:

Reviewer #1 (Remarks to the Author):

The authors have addressed all my comments.

Maybe in proof stage, the Reporting Summary form can be updated with actual GEO database numbers:

" ... RNA-seq data a... have been deposited at the Gene Expression Omnibus, accession number pending."

Reviewer #2 (Remarks to the Author):

The authors responded well to comments, I do not have further comments.

Reviewer #3 (Remarks to the Author)

Funari report use of iPS cells for engineering cancer mutations. They focus on glioblastoma and engineer driver mutations and capture the resulting neural progenitors and use these to generate human GBM models via orthotopic transplantation into mice.

Similar approaches has previously been used by the lab of Vivien Tabar, to model paediatric DIPG, and is also an approach that has been widely used in other organoid systems, such as intestinal organoids, and iPS derived neural differentiation models. Engineering of mutations directly into primary human stem cells has therefore been widely used now and novelty is therefore lost a little bit. See review by Tuveson and Clevers in Science 2019.

However, novelty is not an issue as they have gone on to use this system to demonstrate new insights into glioblastoma biology and genetic evolution. I believe they succeed in doing this, and they are correct in highlighting the importance of generating an isogenic genetic/cellular series for studying the ethology of GBM.

Particularly noteworthy is the emergence of double minute chromosomes. This is novel and interesting. It is surprising that they did not highlight this more in the abstract and title. I think this is significant advance and should enable mechanistic understanding of how these emerge.

A minor criticism. The iPS cell approach seems a bit of a longer process when could have directly engineered human fetal NS cells (which are easily expandable and editable). It is surprising that they chose to engineer the mutations in the iPSCs rather than directly in the neural stem cells; which are not a relevant cell of origin for GBM. They should maybe highlight this as a disadvantage.

It would have been better to expand transformed neural progenitors in vitro with EGF/FGF. Why go back to the iPS cells every time? This is cumbersome, adds weeks to the procedure, and risks contaminating with diverse differentiating cells.

Minor point: Bressan et al., (Development 2017; PMID: 28096221) showed primary human fetal neural stem cells can be engineered with p53 and pH3.3 mutations and they should cite this as the first genome editing in human neural stem cells (although these were not used to produce tumour models).

Also it would be interesting to know if the mesenchymal signatures of the iHGG cells differ before

and after transplantation? This is important to draw conclusions about NF1 loss directly driving this transcriptional signature; alternatively it may have been acquired in vivo following some inductive signals. i.e does the mesenchymal signature require exposure to an in vivo environment?

In summary, this is a high quality study that adds important insights and complementary models to the GBM field.

Reviewer #1 (Remarks to the Author):

The authors have addressed all my comments.

Maybe in proof stage, the Reporting Summary form can be updated with actual GEO database numbers:

" ... RNA-seq data a... have been deposited at the Gene Expression Omnibus, accession number pending."

We appreciate the reviewer's comments. GEO accession numbers were added to the manuscript.

Reviewer #2 (Remarks to the Author):

The authors responded well to comments, I do not have further comments.

We very much appreciate this comment.

Reviewer #3 (Remarks to the Author)

Furnari report use of iPS cells for engineering cancer mutations. They focus on glioblastoma and engineer driver mutations and capture the resulting neural progenitors and use these to generate human GBM models via orthotopic transplantation into mice.

Similar approaches has previously been used by the lab of Vivien Tabar, to model paediatric DIPG, and is also an approach that has been widely used in other organoid systems, such as intestinal organoids, and iPS derived neural differentiation models. Engineering of mutations directly into primary human stem cells has therefore been widely used now and novelty is therefore lost a little bit. See review by Tuveson and Clevers in Science 2019.

We agree that similar approaches to the one we took in this study have already been demonstrated for the modeling of different types of cancers. Recognizing the broad contribution of Dr. Clevers's group to this field, we cited their 2015 Nature paper in the original manuscript. To emphasize the importance of this approach, we now include the following text and citation of their review in Science 2019, as suggested by the reviewer 3.

“These organoid models accurately predict drug responses and their utility is anticipated for application of personalized therapies (Tuveson D, Clevers H, Science 2019).”

However, novelty is not an issue as they have gone on to use this system to demonstrate new insights into glioblastoma biology and genetic evolution. I believe they succeed in doing this, and they are correct in highlighting the importance of generating an isogenic genetic/cellular series for studying the ethology of GBM.

We appreciate the reviewer for recognizing the contribution of our approach to the brain tumor research field.

Particularly noteworthy is the emergence of double minute chromosomes. This is novel and interesting. It is surprising that they did not highlight this more in the abstract and title. I think this is significant advance and should enable mechanistic understanding of how these emerge.

We agree the formation of double minute chromosomes in one of our models was quite intriguing and to the best of our knowledge this was the first example illustrating generation of double minutes in a human stem cell derived model. This was highlighted in the discussion section in the original manuscript. Due to the word limit of the abstract, we were not able to fully describe this finding, but have now added the following description as follows.

“Similar to patient-derived GBM, these models harbor inter-tumor heterogeneity resembling different GBM molecular subtypes, intra-tumor heterogeneity, and extrachromosomal DNA amplification.”

A minor criticism. The iPSC cell approach seems a bit of a longer process when could have directly engineered human fetal NS cells (which are easily expandable and editable). It is surprising that they chose to engineer the mutations in the iPSCs rather than directly in the neural stem cells; which are not a relevant cell of origin for GBM. They should maybe highlight this as a disadvantage.

We agree that direct genome engineering in NS cells would be a potential option for modeling of GBM. One of the reasons we started from iPSCs was because we were originally aiming to model different tumor types, not only brain tumors, using this iPSC-based platform. To clarify this point, the following sentences were added to the discussion section.

“One limitation in our approach is that genome engineering was performed in hiPSCs, an irrelevant cell of origin for GBM. However, the fact that tumor models derived from appropriately differentiated NPCs from edited hiPSCs recapitulate GBM pathobiology suggests that our platform has potential for broader cancer modeling using various cell lineage differentiation protocols applied to hiPSCs.”

It would have been better to expand transformed neural progenitors in vitro with EGF/FGF. Why go back to the iPSC cells every time? This is cumbersome, adds weeks to the procedure, and risks contaminating with diverse differentiating cells.

We apologize for not explicitly describing that point. Our approach was to differentiate genome engineered iPSCs to neural progenitor cells and expanded them in maintenance media described by Reinhardt and colleagues (Reinhardt, P et al., PLoS One, 2013). Once we obtained edited neural progenitors, we did not go back to the iPSCs. To clarify this point, the following sentence was added to the Results section.

“These edited NPCs were expanded on matrigel coated plates in NPC maintenance media (Reinhardt, P et al., PLoS One, 2013) and were utilized in further experiments.”

Minor point: Bressan et al., (Development 2017; PMID: 28096221) showed primary human fetal neural stem cells can be engineered with p53 and p3.3 mutations and they should cite this as the first genome editing in human neural stem cells (although these were not used to produce tumour models).

We thank the reviewer for pointing out this important paper. We cited this paper in the Introduction section and added the following sentence.

“Such engineering has also been efficiently applied to neural stem cells providing opportunities for functional genetic analysis.”

Also it would be interesting to know if the mesenchymal signatures of the iHGG cells differ before and after transplantation? This is important to draw conclusions about NF1 loss directly driving this transcriptional signature; alternatively it may have been

acquired in vivo following some inductive signals. i.e does the mesenchymal signature require exposure to an in vivo environment?

The NPCs with those gene edits did not present any subtype specific transcriptional signature as shown in supplementary figure 7. We speculate that some processes of transformation, either associated with or without microenvironment, might be necessary for those mutant cells to acquire such subtype specific signatures. The following sentence was added to the discussion section.

“Also, how the NPCs with GBM associated mutations, which prior to orthotopic engraftment do not show GBM subtype specific transcriptome signatures, present such signatures through the process of in vivo transformation, is to be further investigated.”

In summary, this is a high quality study that adds important insights and complementary models to the GBM field.

We very much appreciate the reviewer's supportive comment.